# Habitat-Matterport 3D Dataset (HM3D):
# 1000 Large-scale 3D Environments for Embodied AI

**Santhosh K. Ramakrishnan**[1,2]**, Aaron Gokaslan**[1,5]**, Erik Wijmans**[1,3]**, Oleksandr Maksymets**[1]**,
Alex Clegg**[1]**, John Turner**[1]**, Eric Undersander**[1]**, Wojciech Galuba**[1]**, Andrew Westbury**[1]**,
Angel X. Chang**[4]**, Manolis Savva**[4]**, Yili Zhao**[1]**, Dhruv Batra**[1,3]

[1]Facebook AI Research  [2]UT Austin  [3]Georgia Tech  [4]Simon Fraser University  [5]Cornell University

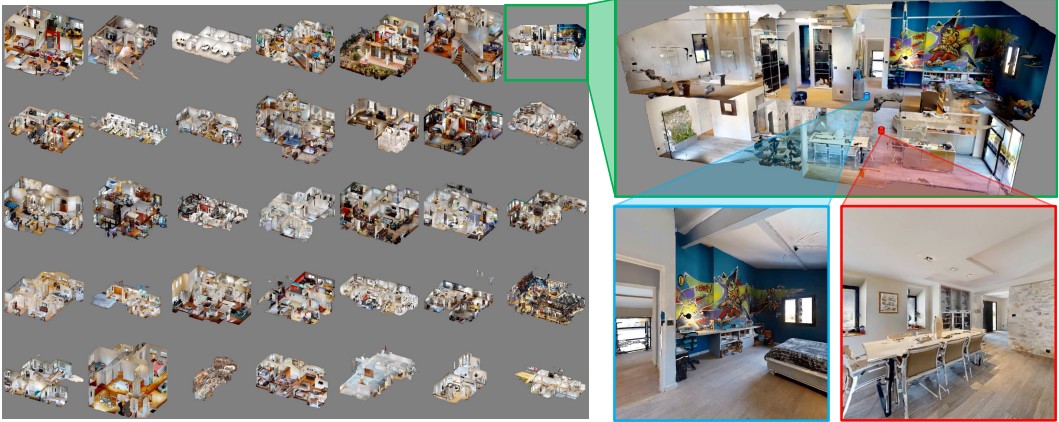

**Figure 1:** The Habitat-Matterport 3D (HM3D) dataset of large-scale 3D and photorealistic environments provides 1,000 building-scale reconstructions of interiors from a diverse set of geographic locations. The scale, completeness, and visual fidelity of these reconstructions surpass those of prior datasets, and enable research on embodied AI agents that can perceive, navigate, and act within realistic indoor environments. The image on the left displays a collage of a subset of HM3D scans. The image on the top-right is a close-up view of a specific scan, and the images on the bottom-right are snapshots from two camera viewpoints in the scan.

## Abstract

We present the Habitat-Matterport 3D (HM3D) dataset. HM3D is a large-scale dataset of 1,000 building-scale 3D reconstructions from a diverse set of real-world locations. Each scene in the dataset consists of a textured 3D mesh reconstruction of interiors such as multi-floor residences, stores, and other private indoor spaces.

HM3D surpasses existing datasets available for academic research in terms of physical scale, completeness of the reconstruction, and visual fidelity. HM3D contains $112.5k$ m$^2$ of navigable space, which is 1.4 - 3.7$\times$ larger than other building-scale datasets such as MP3D and Gibson. When compared to existing photorealistic 3D datasets such as Replica, MP3D, Gibson, and ScanNet, images rendered from HM3D have 20 - 85% higher visual fidelity w.r.t. counterpart images captured with real cameras, and HM3D meshes have 34 - 91% fewer artifacts due to incomplete surface reconstruction.

The increased scale, fidelity, and diversity of HM3D directly impacts the performance of embodied AI agents trained using it. In fact, we find that HM3D is 'pareto optimal' in the following sense – agents trained to perform PointGoal navigation on HM3D achieve the highest performance regardless of whether they are evaluated on HM3D, Gibson, or MP3D. No similar claim can be made about training on

other datasets. HM3D-trained PointNav agents achieve $100\%$ performance on Gibson-test dataset, suggesting that it might be time to retire that episode dataset. The HM3D dataset, analysis code, and pre-trained models are publicly released: https://aihabitat.org/datasets/hm3d/.

# 1 Introduction

As we seek to develop intelligent AI agents that can assist us in our daily activities, good models of indoor 3D environments are becoming increasingly important. Consequently, recent years have seen growing demand for datasets of 3D interiors, whether acquired from the real world, or authored by artists using 3D design tools. Scene datasets based on real-world interiors can be used to develop and evaluate computer vision systems (e.g., on object detection and semantic segmentation tasks), or to train AI agents to navigate and follow instructions in an embodied setting. The latter research agenda in particular has been accelerated by the availability of realistic 3D datasets and high-performance simulators that dramatically reduce the time and logistical complexity for developing AI agents.

Unfortunately, there are only a handful of datasets of indoor 3D environments captured from the real world. Early efforts on 3D scene datasets such as SceneNN [1] and ScanNet [2] collected reconstructions of regions of rooms, and individual rooms. Other datasets that provide 3D reconstructions of entire buildings such as the BuildingParser [3], Matterport3D [4] and Gibson [5] efforts are either limited in total size or suffer from incomplete reconstructions.

We present the Habitat-Matterport 3D Dataset (HM3D), a large dataset of building-scale reconstructions of a diverse set of real-world spaces. HM3D provides $1,000$ near-complete high-fidelity reconstructions of entire buildings (see Figure 1). Each of these reconstructions provides a capture of the habitable and navigable space of each interior. In total, the dataset contains more than $10,600$ rooms across approximately $1,920$ building floors with a navigable area of $112.5k$ m$^2$. The real-world interiors from which these reconstructions are acquired span a diverse set of categories (eg. multi-floor residences, offices, restaurants, and shops), geographical locations, and physical sizes.

Three key characteristics distinguish HM3D relative to prior work on real-world scanned indoor 3D datasets: *scale*, *completeness*, and *visual fidelity*. Unlike prior datasets, each scene in HM3D typically represents a complete building such as a multi-floor private residence. Therefore, HM3D has significantly higher total navigable area ($1.4$ - $3.7\times$ larger), which is particularly important for embodied AI tasks such as navigation. The completeness of HM3D is reflected in $34$ - $91\%$ reduction in reconstruction artifacts due to missing surfaces, holes, or untextured surface regions when compared to prior photorealistic 3D datasets. This increased surface completeness leads to lower incidence of highly unrealistic 'seeing through a hole in the wall' issues that can be detrimental to embodied AI agent training. Finally, the visual fidelity of images rendered from HM3D is $20$ - $85\%$ higher than prior large-scale datasets, which can help to train better embodied AI agents that generalize to real-world settings. As the name suggests, HM3D is 'Habitat-ready', meaning that it comes prepacked with meta-data and support necessary to be used with the Habitat simulator [6] for training embodied AI agents to understand and navigate 3D spaces.

We carry out a number quantitative analyses and experiments to understand the characteristics of HM3D. First, we compare rendered images from HM3D and other 3D scan datasets to camera-captured images from the counterpart real-world interiors, and find that HM3D has significantly higher visual fidelity than other datasets. Second, we find that HM3D has fewer artifacts leading to incompleteness and 'holes' in surface reconstruction. Finally, we train agents for the task of PointGoal navigation [7] using HM3D and other datasets, and find that agents trained in HM3D generalize well across environments. In particular, HM3D is pareto-optimal in the sense that HM3D-trained agents achieve the best performance across Gibson, MP3D, and HM3D test sets. HM3D-trained agents also achieve perfect success on Gibson test scenes, and obtain $3$ - $4$ points higher success and SPL on MP3D test scenes when compared to the next-best agent. These results strongly suggest that embodied agents benefit from the increased scale and diversity of HM3D.

# 2 Related Work

3D datasets can broadly be categorized into synthetic/CAD-based, 3D reconstruction or mesh-based, floorplan-based and panorama-based datasets.

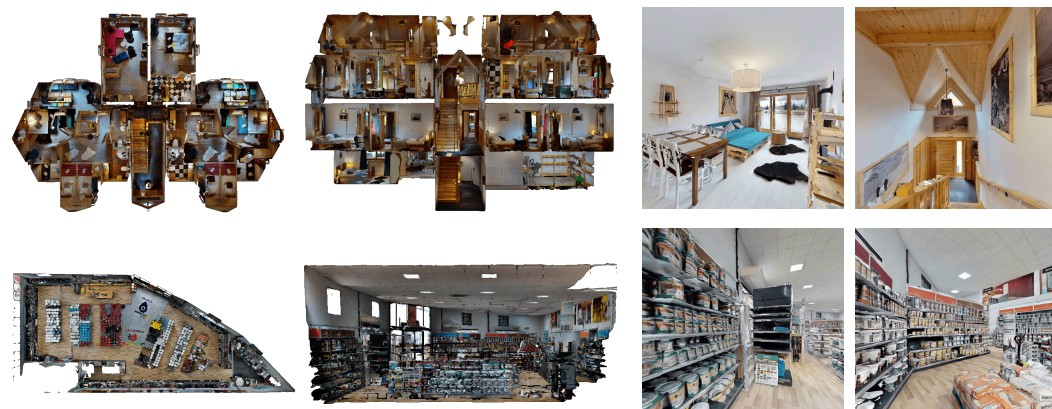

Figure 2: Two example scenes from the HM3D dataset. From left to right in each row: top-down view, cross section view, and two egocentric views from navigable positions in the scene. The dataset contains a wide range of environments such as residences, stores, and workplaces. See Supplementary Section S9 for more examples.

**Synthetic 3D scene datasets.** Embodied AI simulation engines often make use of synthetic scenes with rearrangeable objects [8–11]. Often these scenes are limited to isolated rooms or individual rooms that are connected via a magic portal [9]. There are also datasets of building-scale synthetic scenes [11–13]. However, these authored scenes often do not reflect the variety of architectural layout as well as object arrangement and clutter in the real world. Typically, objects in datasets of synthetic scenes are limited in visual and geometric diversity, since the same set of objects are reused across scenes. In addition, there is a sim-to-real gap between the rendered appearance of synthetic objects and real-world objects. To limit this discrepancy between synthetic and real domains, there are a number of recent synthetic scene dataset efforts that are designed from real world counterpart environments [11, 14, 15]. HM3D is a reconstruction dataset capturing the layout and appearance of a large number of real buildings.

**3D reconstruction datasets.** Existing reconstructions of indoor spaces are limited in scale. Common reconstruction datasets consist primarily of scans for regions of rooms and single rooms [1, 2, 16–18].[1] There exist datasets with building level reconstruction, but these are limited in the overall number of scenes and real-world spaces (BuildingParser [19], 2D-3D-S [3], Matterport3D [4]). The largest building-level reconstruction dataset is Gibson [20] which consists of 571 scenes. However, many scans from Gibson suffer from reconstruction artifacts and 'holes' due to partially reconstructed surfaces. Prior work performed manual inspection and found that only 106 / 571 Gibson scans are of acceptable quality (i.e., $\geq 4$ on a scale of 0-5) [6]. Dehghan et al. [21] have recently released reconstructions of 1,661 scenes but most are single room-scale regions, from rented homes in three European cities. HM3D contains 1,000 building-scale reconstructions spanning a diverse set of locations around the world.

**Floorplan and panorama datasets.** Floorplan datasets [22–24] can be converted to 3D floorplans outlining the architectural layout of buildings and rooms using heuristics. However, the architectural layout tends to be oversimplified as there is typically no specification of wall height or ground level (i.e. all rooms have equal height and simple flat ceilings). Most importantly, these datasets do not provide the textured appearance of the environments or of furniture and other objects present in the rooms. Recently, the Zillow indoor dataset [25] provides floorplans and panorama images captured from a variety of properties. However, almost all captured properties are unfurnished, and even with panorama images available it is not easy to produce 3D mesh reconstructions of the interiors. In contrast, HM3D provides a large number of interiors that are reconstructed to a higher surface completeness and with higher visual fidelity than prior reconstruction datasets.

---

[1]ScanNet and Replica do contain a number of multi-room scenes.

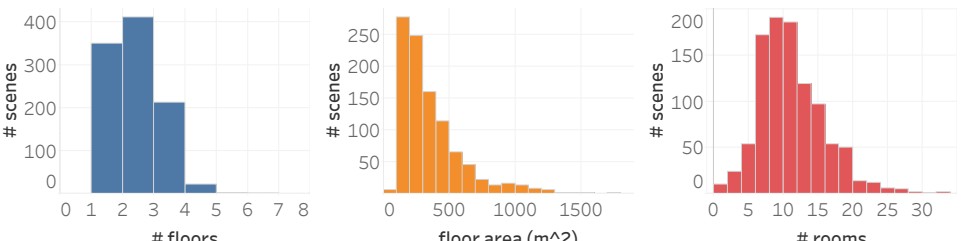

Figure 3: From left to right: i) histogram of distribution over number of floors per scene; ii) histogram of total floor area; iii) histogram of distribution over number of rooms per scene. HM3D scenes span a broad spectrum of physical scale.

## 3   Dataset

The Habitat-Matterport 3D Dataset (HM3D) is a collection of 1,000 3D reconstructions and consists of multi-floor residences, stores, and other private indoor spaces. All spaces were scanned using a Matterport Pro2 tripod-based depth sensor.[2] Alignment of the RGB-D data, surface meshing, and texturing were carried out using the reconstruction pipeline provided by Matterport, Inc. Scans were taken from spaces in 38 countries, and 181 geographic regions (states, provinces etc.) across those countries. In the United States, spaces are located in 43 states. Figure 2 shows some example scenes.

The set of 1,000 scenes in HM3D were curated from a larger pool of candidate scenes through a two-stage annotation and verification process. First, a group of 15 volunteer annotators rated each scene on a 1-5 quality scale assessing scene quality. The annotators visually inspected each scene for reconstruction artifacts such as holes/cracks, the presence of realistic and dense furnishing, the number of closed doors (which prevent access to some rooms), and the 'interactive potential' of the scene (based on objects with which a person might interact). The annotators also had access to quantitative metrics characterizing the navigability of the scene so that they could detect reconstruction issues causing disconnectedness in the building floors. After a round of rating, a second set of three expert annotators collated scenes by first sorting highly ranked scenes and then selecting so as to preserve diversity in the number of floors per scene. In total, HM3D curation, annotation, and release represents an estimated 800+ hours of human effort.

The final set of scenes spans a broad spectrum of total area, with the smallest scene having a floor area of $49m^2$ and the largest scene an area of $2,172m^2$. The architectural layout of the scenes also spans a broad spectrum with buildings of between one and eight floors, and between one 'room' and 93 rooms.[3] More detailed statistics regarding the dataset composition are visualized in Figure 3.

### 3.1   Scale comparison

We compare the scale of HM3D to other datasets using a number of metrics that measure the overall floor area, navigable area, and structural complexity of the scenes.

**Floor area** ($m^2$) measures the overall extents of the floor regions in the scene. This is the area of the 2D convex hull of all navigable locations in a floor. For scenes with multiple floors, the floor space is summed over all floors. This is implemented in the same way as by Xia et al. [5] to make the reported statistics comparable. Higher values indicate the presence of more navigation space and rooms.

**Navigable area** ($m^2$) measures the total scene area that is actually navigable in the scene. This is computed for a cylindrical robot with radius 0.1m and height 1.5m using the AI Habitat [6] navigation mesh implementation. This area is strictly lower than the floor area as it excludes points that are not reachable by the robot. Higher values indicate larger quantity and diversity of viewpoints for a robot.

**Navigation complexity** measures the difficulty of navigating in a scene. This is computed as the maximum ratio of geodesic path to euclidean distances between any two navigable locations in the scene. This is the same metric as reported for the original Gibson dataset to again make the statistics

---

[2]https://matterport.com/cameras/pro2-3D-camera

[3]Room statistics were obtained using the mesh chunk meta-data from the Matterport reconstruction pipeline. Each mesh chunk is created by the reconstruction pipeline from a set of tripod locations in the same room.

| Dataset | Replica [16] | RoboTHOR [14] | MP3D [4] | Gibson [5] (4+ only) | ScanNet [2] | HM3D (ours) |
|---|---|---|---|---|---|---|
| Number of scenes | 18 | 75 | 90 | 571 (106) | 1613 | 1000 |
| Floor area (m$^2$) | 2.19$k$ | 3.17$k$ | 101.82$k$ | 217.99$k$ (17.74$k$) | 39.98$k$ | 365.42$k$ |
| Navigable area (m$^2$) | 0.56$k$ | 0.75$k$ | 30.22$k$ | 81.84$k$ (7.18$k$) | 10.52$k$ | 112.50$k$ |
| Navigation complexity | 5.99 | 2.06 | 17.09 | 14.25 (11.90) | 3.78 | 13.31 |
| Scene clutter | 3.4 | 8.2 | 2.99 | 3.14 (3.04) | 3.15 | 3.90 |

Table 1: Comparison of HM3D to other existing indoor scene datasets. Gibson 4+ refers to the subset of Gibson scenes that were rated as "high quality" and relatively free of reconstruction errors [6]. HM3D surpasses previously available datasets with $1.8\times$ more scans, 1.6 - $3.6\times$ higher total physical size, provides $1.2\times$ more cluttered scenes, and has relatively high navigation complexity.

comparable [5]. Higher values indicate more complex layouts with navigation paths that deviate significantly from straight-line paths.

**Scene clutter** measures the amount of clutter in the scene. This is computed as the ratio between the raw scene mesh area within $0.5$m of the navigable regions and the navigable space. We restrict to $0.5$m to only pick the surfaces that are near navigable spaces in the building (e.g., furniture, and interior walls), and to ignore other surfaces outside the building. Higher values are better and indicate more cluttered scenes that provide more obstacles for navigation.

Table 1 reports the values of these metrics for HM3D as well as a number of other indoor datasets, primarily focusing on existing 3D reconstruction datasets. We also compute the metrics for the RoboTHOR [14] dataset which is synthetic but based on real-world layouts. The chosen comparison points span a spectrum of total sizes and complexities. For Gibson, note a second set of metric values for the restricted subset of fewer "high quality" Gibson scenes that were rated as at least 4/5 by a set of human annotators. This subset of Gibson exhibits fewer reconstruction artifacts than the full Gibson dataset (see Savva et al. [6] for a description of the original rating process).

We can make a number of observations. First, HM3D provides $1.7\times$ higher floor area and $1.4\times$ higher navigable area compared to the Gibson (previously the largest). In particular, HM3D provides $20\times$ higher floor area and $15.6\times$ higher navigable area if we only consider the high quality reconstructions in Gibson 4+. Second, the scene clutter of HM3D exceeds that of most other datasets by $\sim 1.2\times$ with the exception of RoboTHOR which is a significantly smaller dataset. Finally, the navigation complexity metric shows that HM3D scenes are relatively complex to navigate, close to other building-scale datasets such as MP3D and Gibson (by a factor of 0.8 - $1.1\times$), and higher than room-scale datasets such as Replica, RoboTHOR and ScanNet (by a factor of 2.2 - $6.4\times$).

### 3.2 Reconstruction completeness comparison

When reconstructing a 3D mesh from scanned images, it is common to encounter reconstruction artifacts (or defects) such as missing surfaces, holes, or untextured surface regions. These reconstruction artifacts lower the visual quality of rendered images and may increase the domain gap between learning within the simulator and deploying to the real world. HM3D offers more complete reconstructions with fewer instances of missing surfaces or reconstruction artifacts resulting in 'holes' and 'black cracks'. We design a view-based metric to measure the degree to which such artifacts occur in HM3D and other datasets. First, we densely sample camera viewpoints in a scene as follows. We divide the set of navigable locations in a scene into a grid with $1$m $\times$ $1$m cells. At each grid location, we sample 4 camera viewpoints by varying the agent's heading angle ($0°, 90°, 180°, 270°$) and fixing the azimuth angle to $0°$ (i.e., agent looks straight ahead). Finally, we place a $90°$ field-of-view RGB-D camera at each viewpoint and render $256 \times 256$ images[4] with a camera height of $1.5$m using `habitat-sim 0.1.7` [6]. This sampling process is uniform (in terms of coverage of the floor space) and adaptive (i.e., the number of images varies with the size of the scene). For each viewpoint, we compute the fraction of depth pixels with depth value of $0$ (i.e., invalid depth pixels) and the fraction of RGB pixels that are completely black (i.e. RGB value of $0$).[5] Any views with more than 5% of such pixels either in the depth frame or the RGB frame are marked as exhibiting an artifact. Figure 4 (left) shows examples of views with significant defects. In reconstructions that are complete and that do not exhibit holes and cracks, there will be fewer such views. We summarize this metric by

---

[4]All the datasets we compare with already have meshes and textures available.

[5]The pixels corresponding to missing surfaces / textures are set to 0 for both RGB and depth rendering.

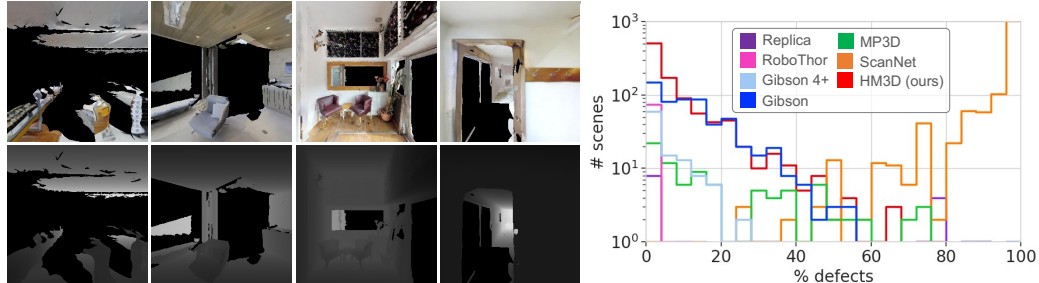

Figure 4: We measure the incidence of reconstruction artifacts such as missing surfaces, holes, or untextured surface regions ('defects') using a view-based metric. **Left:** Examples of viewpoints with significant reconstruction artifacts. **Right:** Histogram of defect metric over the scenes from different datasets. The X-axis measures the fraction of densely sampled views in a scene that exhibit significant defects. The Y-axis is the number of scenes (in log-scale). HM3D provides the largest number of scans with minimal reconstruction artifacts.

computing the proportion of sampled views in a scene that exhibit such significant reconstruction artifacts to arrive at an overall '% defects' value for the scene.

We compare the distribution of this metric for HM3D and other datasets in Figure 4 (right). Overall, we see that HM3D scenes exhibit fewer artifacts (more scenes with lower '% defect' values). While ScanNet offers more scans than HM3D, almost all scans from ScanNet exhibit severe reconstruction artifacts. Other large-scale datasets such as Gibson, and MP3D exhibit broader distributions with a significant number of scenes having fairly high reconstruction defect values. HM3D has more than three times as many scenes with less than 5% of views exhibiting artifacts compared to Gibson (560 scenes vs 175 scenes). As expected, Gibson 4+ provides a smaller but higher-quality subset of Gibson scenes with fewer reconstruction artifacts. While RoboTHOR scans have very few artifacts, they are not photorealistic and are small in quantity. The Replica scans are much smaller in number (only 18 scenes), and 6 / 18 scans have $\sim 80\%$ defects since the roof is missing. Overall, HM3D offers the largest number of scans with high completeness.

## 3.3 Visual fidelity comparison

We also compare the overall visual quality of rendered images from HM3D with prior datasets. For each dataset, we use the RGB images from Section 3.2 to ensure that we assess the visual fidelity of rendered images from all parts of a scene. We compare the image quality against a set of real RGB images generated from high-resolution panoramas (i.e., 360° field-of-view equirectangular images) in Gibson and MP3D using the FID [26] and KID [27] metrics. We refer to these sets of real RGB images as 'Gibson real' and 'MP3D real'. Gibson real has 124,227 images sampled from 41,409 panoramas. MP3D real has 98,208 images sampled from 10,912 panoramas. Figure 5 summarizes the results of this comparison.[6]

The quality of images rendered from HM3D is much closer to real images when compared to the other datasets. Out of all datasets, images rendered from HM3D exhibit the lowest FID / KID scores when compared with both MP3D real (20.53/15.78) and Gibson real (20.49/12.76). Note that we observe a domain shift between datasets that leads to non-zero FID and KID scores even for real images. Comparing images from Gibson real with MP3D real provides a 'lower bound' of $6.16 - 6.23$ against which we can compare the metric values for rendered dataset images. As expected, images rendered from RoboTHOR have the highest FID and KID scores since they are not photorealistic. Images rendered from ScanNet also have high FID and KID scores since they exhibit significant mesh artifacts (see Sec. 3.2). Images rendered from Replica have relatively high distance scores (despite having high quality scans) due to the lack of textured ceilings in multiple scans (eg., 34.94 FID vs. Gibson real, and 42.76 FID / 19.31 KID vs. MP3D real). Images rendered from the remaining datasets have significantly higher FID and KID values when compared to HM3D, showing that they have lower visual fidelity as measured against the real images from Gibson and MP3D. Overall, images rendered from HM3D have 20 - 85% higher visual fidelity relative to existing photorealistic 3D datasets based on the mean KID scores in Figure 5.

---

[6]Figure 5 (a) is illustrative. We approximately matched the pose of real and simulated images.

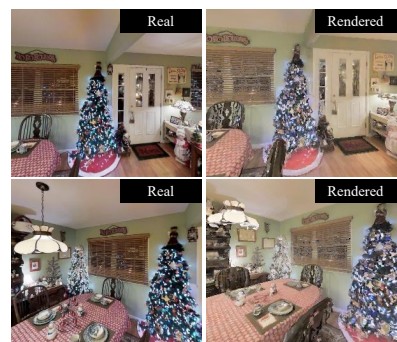

(a) Real vs. rendered images.

| Dataset | # scenes | Gibson real | | MP3D real | |
|---|---|---|---|---|---|
| | | FID ↓ | KID×$10^3$ ↓ | FID ↓ | KID×$10^3$ ↓ |
| Replica | 18 | 34.9 | **15.8** ± 0.8 | 42.8 | 19.3 ± 0.8 |
| RoboTHOR | 75 | 157.6 | 109.8 ± 1.8 | 163.0 | 111.3 ± 1.8 |
| MP3D | 90 | 43.8 | 32.9 ± 1.6 | 24.4 | 17.3 ± 1.2 |
| Gibson 4+ | 106 | 27.4 | 18.9 ± 0.8 | 32.6 | 20.0 ± 0.8 |
| Gibson | 571 | 39.3 | 29.8 ± 1.5 | 38.0 | 25.5 ± 1.0 |
| ScanNet | 1613 | 126.7 | 106.0 ± 2.4 | 121.3 | 97.4 ± 2.30 |
| HM3D | 1000 | **20.5** | **15.8** ± 1.0 | **20.5** | **12.8** ± 0.8 |
| MP3D real | 90 | 11.2 | 6.2 ± 0.7 | 0.0 | 0.0 ± 0.1 |
| Gibson real | 571 | 0.0 | 0.0 ± 0.1 | 11.2 | 6.2 ± 0.7 |

(b) Visual fidelity comparison of rendered images with real images.

Figure 5: Visual fidelity comparison of HM3D and other reconstruction datasets. We render images from the reconstructed scenes in each dataset using Habitat [6], and extract real RGB images from raw panoramas in Gibson and MP3D (see comparison on the left). Then, we compute the FID and KID of the rendered images by comparing them against the real images. Lower values indicate closer distributional match to the statistics of the real image sets. HM3D provides significantly lower FID and KID than images rendered from other datasets, even in the case of images rendered from Gibson reconstructions evaluated against Gibson real images. This result indicates the high visual fidelity of HM3D reconstructions relative to other datasets.

## 4 Experiments

A popular downstream application for large-scale 3D reconstruction datasets has been to use them with 3D simulation platforms [6, 14, 20] to study embodied AI tasks such as visual navigation [6, 28–31]. As described in the previous sections, HM3D improves over existing datasets both in terms of size and quality. In this section, we perform experiments to show that navigation agents trained on HM3D benefit from its scale and quality, and generalize better when transferred to other datasets.

### 4.1 Experimental setup

We benchmark embodied agents on the PointGoal navigation (a.k.a. PointNav) task [7], which has served as a standard testbed for exploring ideas in navigation [30, 32–34] and a starting point for more semantic tasks [35–37]. In PointNav, an agent is randomly spawned inside a novel environment, and is given a navigation goal coordinate $(\Delta x, \Delta y)$ relative to its starting location. It has to efficiently navigate to this goal using visual inputs (RGB or depth sensors). Specifically, we consider the PointNav-v1 task [6], where the agent's observation space consists of $256 \times 256$ RGB-D visual inputs, and GPS+Compass readings for localization. The agent's action space is [MOVE FORWARD, TURN LEFT, TURN RIGHT, STOP]. The forward step size is $0.25$m and the turn angle is $10°$. The agent succeeds if it reaches within $0.2$m of the goal location and executes STOP. We evaluate PointNav performance using (1) Success, which measures the fraction of episodes successfully completed, and (2) SPL, which measures the efficiency of navigation relative to the shortest paths [7].

We train and evaluate PointNav agents on Gibson 4+, Gibson, MP3D, and HM3D datasets. We divide the 1,000 HM3D scenes into disjoint sets of 800 train / 100 val / 100 test scenes. We use the standard train / val / test splits for Gibson 4+ and MP3D [6]. We create new PointNav episode datasets for the full Gibson train scenes and HM3D using the generation script from Savva et al. [6]. Specifically, we generate 10,000 episodes for each train scene, and 25 episodes for each val/test scene. This results in 4.11M train episodes for Gibson, and 8.0M train / 2,500 val / 2,500 test episodes for HM3D. These splits are publicly available to aid reproducibility: https://github.com/facebookresearch/habitat-lab. We compare the difficulties of the validation episode datasets for Gibson, MP3D, and HM3D in Figure 6. In general, MP3D has the hardest episodes and Gibson has the easiest episodes.

We use a standard agent architecture for training on different datasets [30]. A ResNet-50 backbone extracts visual features [38], and an MLP extracts location features from GPS+compass readings. An LSTM state-encoder aggregates these features over time [39], and fully-connected layers are used to predict action logits (i.e., the policy) and state values (i.e., the value function). Actions are then stochastically sampled from the predicted action logits. We train the agent using DD-PPO for 1.5 billion frames which was shown to be sufficient to achieve near state-of-the-art performance [30].

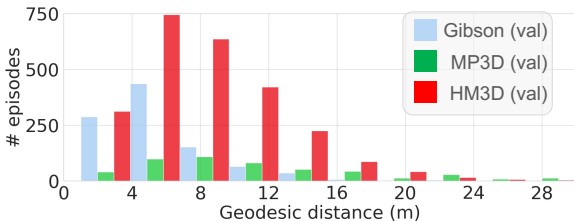

Figure 6: Distribution of PointNav episode difficulties in the val splits of Gibson, MP3D, and HM3D. We group the episodes in each dataset based on the geodesic distance between the start and goal positions. Episodes with larger geodesic distance are generally harder. Gibson has the easiest episodes. MP3D has the hardest episodes.

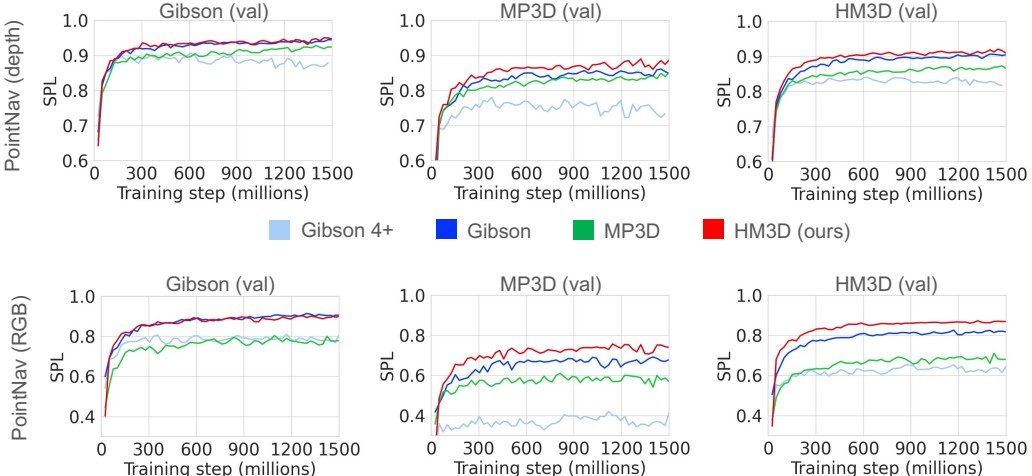

Figure 7: **PointNav validation performance vs. training steps:** The top row shows results with depth-only agents, and the bottom row shows results with RGB-only agents. Training on HM3D leads to faster convergence and better generalization to newer scenes and datasets.

|  | Dataset | Gibson (test) | | MP3D (test) | | HM3D (test) | |
|---|---|---|---|---|---|---|---|
|  |  | Success ↑ | SPL ↑ | Success ↑ | SPL ↑ | Success ↑ | SPL ↑ |
| Depth | MP3D | $0.97 \pm 0.01$ | $0.90 \pm 0.01$ | $0.89 \pm 0.00$ | $0.80 \pm 0.00$ | $0.96 \pm 0.00$ | $0.87 \pm 0.00$ |
| Depth | Gibson 4+ | $0.96 \pm 0.02$ | $0.90 \pm 0.02$ | $0.77 \pm 0.01$ | $0.68 \pm 0.01$ | $0.93 \pm 0.00$ | $0.84 \pm 0.00$ |
| Depth | Gibson | $\mathbf{1.00} \pm 0.00$ | $\mathbf{0.94} \pm 0.01$ | $0.90 \pm 0.00$ | $0.80 \pm 0.00$ | $0.98 \pm 0.00$ | $0.90 \pm 0.00$ |
| Depth | HM3D | $\mathbf{1.00} \pm 0.00$ | $0.93 \pm 0.01$ | $\mathbf{0.94} \pm 0.00$ | $\mathbf{0.83} \pm 0.00$ | $\mathbf{0.99} \pm 0.00$ | $\mathbf{0.92} \pm 0.00$ |
| RGB | MP3D | $0.91 \pm 0.01$ | $0.73 \pm 0.03$ | $0.72 \pm 0.01$ | $0.56 \pm 0.01$ | $0.85 \pm 0.00$ | $0.68 \pm 0.00$ |
| RGB | Gibson 4+ | $0.88 \pm 0.01$ | $0.73 \pm 0.03$ | $0.44 \pm 0.00$ | $0.35 \pm 0.00$ | $0.77 \pm 0.00$ | $0.62 \pm 0.00$ |
| RGB | Gibson | $0.96 \pm 0.00$ | $0.87 \pm 0.01$ | $0.82 \pm 0.01$ | $0.68 \pm 0.01$ | $0.94 \pm 0.00$ | $0.82 \pm 0.00$ |
| RGB | HM3D | $\mathbf{1.00} \pm 0.01$ | $\mathbf{0.90} \pm 0.02$ | $\mathbf{0.85} \pm 0.01$ | $\mathbf{0.71} \pm 0.01$ | $\mathbf{0.98} \pm 0.00$ | $\mathbf{0.87} \pm 0.00$ |

Table 2: **PointNav test performance** on multiple navigation metrics. We report the mean and standard deviation by training on 1 random seed, and evaluating on 3 random seeds. The $1^{st}$ column indicates whether the agent uses depth or RGB inputs. The HM3D agents reach $100\%$ navigation success for both sensors on Gibson (test). In the majority of cases, HM3D agents significantly outperform the other agents on both metrics. Thus, training on HM3D greatly benefits embodied agents.

We separately benchmark agents for two types of inputs. For 'RGB inputs', the agent navigates using RGB and GPS+compass sensors. For 'depth inputs', the agent navigates using depth and GPS+compass sensors. For brevity, 'X agent' refers to an agent trained on dataset X (e.g., Gibson agent), and 'X agent (R, D)' denotes the SPL performance of X agent with RGB (R) and depth (D) inputs. Figure 7 and Table 2 present results for all agents and datasets. We analyze these results next to answer 3 key questions.

**1) Is HM3D beneficial for training PointNav agents?** Consider the validation curves in Figure 7. Both HM3D agents (RGB and depth inputs) converge faster and perform better than corresponding agents trained on other datasets. Specifically, the HM3D agent closely follows the Gibson agent on

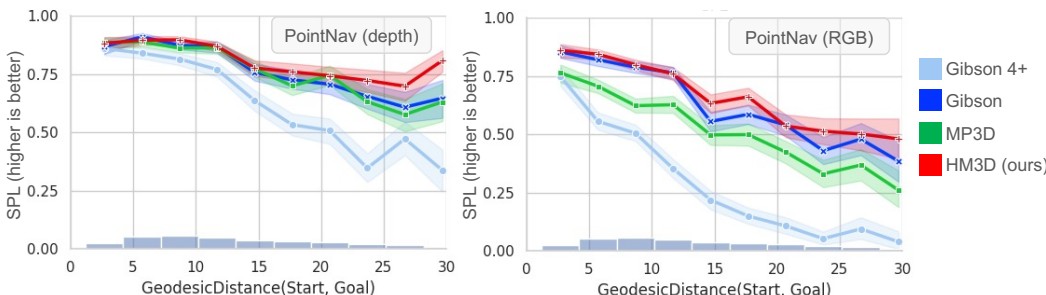

Figure 8: **PointNav SPL vs. episode difficulty:** We divide the MP3D (test) episodes into bins based on the geodesic distance between the start and goal positions (a measure of episode difficulty). For each bin, we report the mean and stddev SPL for PointNav agents with depth (top figure) and RGB (bottom figure) inputs. The diversity and complexity of HM3D layouts allows the HM3D agents to generalize better to harder episodes.

Gibson (val) and outperforms it on MP3D (val) and HM3D (val). The validation performance of the HM3D agent rapidly outpaces the MP3D and Gibson 4+ agents on all cases. The test performance in Table 2 confirms the above trends. On Gibson (test) with depth inputs, the HM3D agent matches the Gibson agent achieving 0.93 SPL and 1.0 success. On all other cases, the HM3D agents outperform the other agents by a large margin. For example, on MP3D (test), the HM3D agent (0.71, 0.83) significantly outperforms the second-best Gibson agent (0.68, 0.80). Thus, HM3D is pareto-optimal since the HM3D agents achieve the best performance on all test sets.

**2) Are HM3D scenes diverse in terms of visual appearance and 3D layouts?** Diversity in visual appearance and 3D layouts in the training scenes is essential for generalization to novel scenes and datasets, and adaptability to difficult PointNav episodes. From Table 2, HM3D agents (both RGB and depth) outperform the next best method on MP3D (test) by 3 SPL points, and achieve perfect success on Gibson (test). This is impressive generalization since the HM3D agent had not observed any Gibson or MP3D scenes during training, and yet was able to overcome the domain gap in appearance and layouts of the scenes. This attests to the visual richness and layout diversity of HM3D which enables good generalization to *previously unseen scenes and datasets*.

Next, we compare the performance of different agents as a function of the episode difficulty in Figure 8. We quantify episode difficulty using the geodesic distance between the start and goal locations [30]. We group the MP3D (test) episodes into different bins based on the above metric, and plot the mean and standard deviation of an agent's performance on all episodes in each bin. We select MP3D (test) since it has highest diversity of difficulty levels (see Figure 6). We observe that the HM3D agent adapts much better compared to other agents as the episode difficulty increases. This is yet another indicator that the layouts in HM3D are *complex and diverse*.

**3) Does PointNav benefit from scaling up 3D datasets?** Prior work has verified the data scaling hypothesis for passive perception, i.e., scaling up the dataset can significantly improve performance on various passive perception problems [40–43]. However, this is not well-established in the embodied perception literature due to the lack of large-scale 3D datasets with high quality. As discussed in earlier sections, HM3D offers large-scale, high visual fidelity, and high-quality reconstructions. Thus, we use HM3D to test the data scaling hypothesis for embodied AI. Specifically, we test the relationship between the training dataset size and the PointNav performance. For this purpose, we additionally train agents on two random subsets of HM3D containing 10% and 50% of the scans in the HM3D train split (i.e., 80 and 400 scans respectively). We refer to these agents as HM3D (10%) and HM3D (50%). Figure 9 shows the PointNav SPL on the test splits as a function of the total navigable area in the training scenes. We observe that the navigation performance is strongly correlated with the total navigation area (Pearson coefficient $\rho = 0.88$), and that the performance scales near-linearly as the total navigable area increases. This result is also helpful to decide the data budget for training PointNav agents. Using more data leads to better performance (particularly on the harder episodes in MP3D), but requires more computational resources and time. Depending on the task difficulty and availability of computational resources, researchers can choose the appropriate dataset(s) for experimentation.

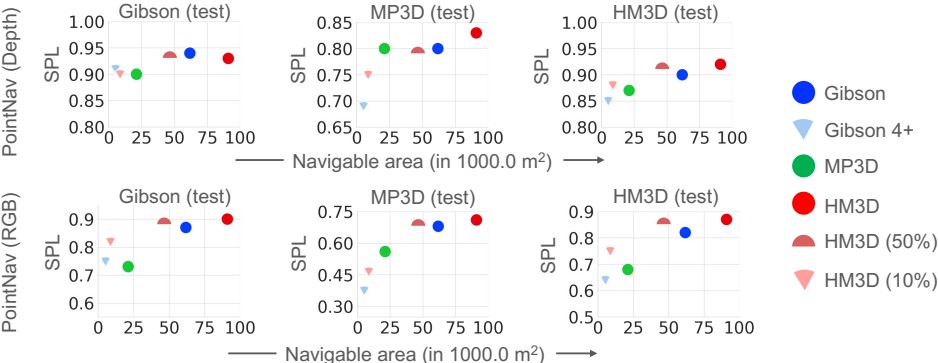

Figure 9: **PointNav test performance vs. navigable area:** PointNav performance scales nearly linearly as a function of the total navigable area in training scans.

## 5 Conclusion

We presented the Habitat-Matterport 3D (HM3D) dataset consisting of 1,000 building-scale reconstructions from the real world. To our knowledge, HM3D offers the largest dataset of high-quality 3D reconstructions of interiors for academic research. Through a series of quantitative analyses we showed that HM3D improves upon existing 3D reconstruction datasets in three ways: significantly larger spatial scale, improved reconstruction completeness, and higher visual fidelity. We also carried out experiments with PointGoal navigation for embodied AI agents to show that agents trained on HM3D match or outperform agents trained on other datasets even when evaluated on other datasets. This demonstrates the value of HM3D as a dataset for embodied AI. Extension of HM3D with object semantics and physical attributes in future work will enable even more embodied AI tasks such as ObjectGoal navigation and object rearrangement. We hope that HM3D will catalyze research in the area of embodied AI.

## 6 Acknowledgements

We thank all the volunteers who contributed to the dataset curation effort: Harsh Agrawal, Sashank Gondala, Rishabh Jain, Shawn Jiang, Yash Kant, Noah Maestre, Yongsen Mao, Abhinav Moudgil, Sonia Raychaudhuri, Ayush Shrivastava, Andrew Szot, Joanne Truong, Madhawa Vidanapathirana, Joel Ye. We thank our collaborators at Matterport for their contributions to the dataset: Conway Chen, Victor Schwartz, Nicole Rogers, Sachal Dhillon, Raghu Munaswamy, Mark Anderson.

## 7 Licenses for referenced datasets

Gibson: http://svl.stanford.edu/gibson2/assets/GDS_agreement.pdf
Matterport3D: http://kaldir.vc.in.tum.de/matterport/MP_TOS.pdf
ScanNet: http://kaldir.vc.in.tum.de/scannet/ScanNet_TOS.pdf
Replica: https://github.com/facebookresearch/Replica-Dataset/blob/master/LICENSE

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
