# OpenReview forum: "Habitat-Matterport 3D Dataset (HM3D): 1000 Large-scale 3D Environments for Embodied AI"
_NeurIPS.cc/2021/Track/Datasets_and_Benchmarks/Round2 — NeurIPS 2021 Datasets and Benchmarks Track (Round 2)_

### Official Review · Reviewer_HSHf · 2021-09-19
**large building-scale and high-quality 3D datasets for indoor navigation, but NO raw images / depths, camera poses / intrinsic and semantic annotation**

**Rating:** 7
**Confidence:** 3
**Correctness:** Put relevant comments in `Clarity` se…

**Strengths:**

- a larger dataset from improved Matterport pipeline (since Matterport3D, 2017), largest building-level 3D datasets, mainly for robot navigation task
- show better performance for point navigation task when trained with this dataset due to its diversity, scale, and perceptual quality

**Weaknesses:**

- only provide the reconstructed mesh and texture, material, but not the raw images / depth, semantic labels, potentially useful for 3d reconstruction and understanding tasks
- only evaluated the datasets for pointnav, but not evaluated for other embodied AI tasks e.g. interaction, manipulation, e.g. compared to iGibson (2.0)

**Additional Feedback:**

see clarity section

**Clarity:**

paragraph around L166
- render images from different views: only mentioned the view angle and (x,y) location (1m x 1m) , what is the height? e.g. human or robot's height? does it matter?
- are the rendering process the same for each dataset? which renderer do you use? (I assume Habitat)
- do you need to reconstruct the mesh and textures for other datasets before rendering novel views, or extract from panoramas like the caption in Fig 5?
- can you render with new lighting conditions? from the navigation video, the shadows are not realistically changing as robot moving around the room

Fig 3
- x labels not quite clear

Fig4 Right
 - is it true there are scenes in ScanNet with 100% defects?
 - maybe better show the y axis in log scale to tell the datasets apart in terms of % defect?
 - How does HM3D compare to Matterport3D for this metric? is the improvement mainly from the improved reconstruction from Matterport's pipeline over the last 4 years?
 - Confused that there are 9 curves shown in the figure but only 7 legends if I look at the left most column?
 - How much does the missing roof contribute to this metric?

Fig5
- Left figure: it is not clear what are the 4 images shown on the left.
- How many real images in Gibson real and MP3D real to compute FID against?
- what is your opinion on the visual quality of rendered images? e.g. compared to GAN literature, are the FIDs score still high, is it good enough for embodied AI tasks? (potential more advanced task than pointnav)

how to do count the area of stairs in multi-floor buildings? how are they used in nav tasks? are the robots capable of going up and down the stairs?

L217 - when evaluate agents (trained on different datasets) on a test environment, does the agents start from the same `random` locations to the same target goal?

L224 - what is SPL?

L230 - the training episodes are 4.11M for Gibson and 8M for HM3D. Is the comparison fair?

Fig 8 - top left subfigure - how do you explain for depth input, when tested on Gibson (test): agents trained on Gibson is better than trained on HM3D even with less training episodes / training dataset size?

**Documentation:**

- dataset has well been released under a friendly licence
- the procedure to access the dataset is reasonable
- the code is also included in the sup material to reproduce the experiment
- sup materials contains details regarding dataset creation, experiments and limitations of current dataset


**Ethics:**

- the datasets are from Matterport users who agree to release for research
- paper also discuss potential bias due to the users who provide the dataset have to be the ones who can afford the Matterport capture device
- geo locations of the buildings are pretty diverse

**Relation To Prior Work:**

- how does this dataset compared to ARKitScenes? both claim to be the largest 3D dataset. It seems ARKitScenes provide all captured images / HR and LR depths, annotations which are missing in this dataset
- missing reference on OpenRooms (https://arxiv.org/abs/2007.12868v2)

**Summary And Contributions:**

This paper presents a new large-scale 3D building-level indoor dataset captured and reconstructed with Matterport's pipeline, mainly used in simulation environment for embodied AI, such as RL-based robot navigation in the room. The dataset is shown to be the largest scale (navigation area), less reconstruction artifacts (e.g. more complete mesh) and high perceptual quality (of rendered images). They also show agents trained in the environment simulated with HM3D achieves better PointNav performance than previous datasets.

---

> ### Author Response · Authors · 2021-09-28
> **Response to reviewer HSHf [Part 1]**
>
> ### Q1 - only evaluated for pointnav, but not other embodied AI tasks e.g. interaction, manipulation
> Since HM3D contains static scans, it does not currently support interaction or manipulation tasks (e.g. opening a cabinet, picking up an object, etc). For a discussion on this topic and conversion of 3D scans to CAD models, please see Section 3 in Szot et al., 2021 (https://arxiv.org/abs/2106.14405). We describe this as a limitation of HM3D in Supplementary Section S1. However, other navigation tasks such as ImageNav, Exploration, ObjectNav with inserted objects are already supported on HM3D (please see response to Q3 from R-LQ43 for task descriptions).
> Since this paper is focused on the HM3D reconstructions, we evaluate on PointNav, a fundamental task that can serve as a “building block” for other embodied AI tasks, thus making it a good starting point (L215-217). Our experiments already indicate that the diversity, scale, and fidelity of HM3D can benefit embodied agents.
>
>
> ### Q2 - Rendering images for visual fidelity evaluation
> We load all the scenes in habitat-sim 0.1.7 for rendering images.
> All other datasets we compare to in the paper (Gibson, MP3D, Replica, ScanNet) already have meshes and textures available, so we do not need to perform any reconstructions ourselves. The panoramas are used only for obtaining the Gibson real and MP3D real images used for visual fidelity comparisons in Section 3.3.
> We use the default height of 1.5m from habitat-sim. Since the same height is used for rendering images from all the datasets, we don’t expect this to affect our analysis.
> The lighting is baked into the reconstructed scenes for HM3D. It is possible to insert point light and environment maps in the habitat-sim renderer (please see https://aihabitat.org/docs/habitat-sim/lighting-setups.html). However, we did not investigate or use them for HM3D.
>
> ### Q3 - x labels for Figure (3) is not clear
> Thank you. We have fixed this.
>
> ### Q4 - Figure 4 right
> * **is it true there are scenes in ScanNet with 100% defects?**
> Yes. Recall L170-4, 100% defects implies 100% of viewpoints sampled from the scan have at least 5% defective RGB/depth pixels.
>
> * **maybe better show the y axis in log scale to tell the datasets apart in terms of % defect?**
> Thank you for the suggestion, we have included this in the paper.
>
> * **How does HM3D compare to Matterport3D for this metric? is the improvement mainly from the improved reconstruction from Matterport's pipeline?**
> Averaging the % defects value in Figure 4 over all scenes within each dataset, we get $8.4$% for HM3D and $22.0$% for MP3D. The improvements are likely due to a combination of improvements in the Matterport reconstruction pipeline and the quality verification process we used to select scenes from a larger candidate pool.
>
> * **9 curves shown in the figure but only 7 legends?**
> The 2 additional curves correspond to HM3D (50%) and HM3D (10%). We missed removing them from the final curves. We have fixed this. Thank you.
>
> *  **How much does the missing roof contribute to this metric?**
> 6 / 18 Replica scenes are missing a roof (frl_apartment_*). These scenes score $75$% - $85$% on the %defects metric. We have clarified this in the paper.
>
> ### Q6 - Figure 5
> * **Left figure: it is not clear what are the 4 images shown on the left.**
> The images on the left column are samples from the raw MP3D images (not rendered from scans). The images on the right column are rendered from the MP3D scans using habitat-sim (from approximately similar viewpoints).
>
> * **How many real images in Gibson real and MP3D real to compute FID against?**
> Gibson real has 124,227 images (3 images for each of 41,409 panorama). MP3D real has 98,208 images (9 images for each of 10,912 panoramas). We have clarified this in the paper.
>
> * **What is your opinion on the visual quality of rendered images? Is it good enough for embodied AI tasks?**
> We believe that the rendered quality is much better than prior datasets thanks to better visual fidelity as well as fewer reconstruction artifacts. We expect this to benefit complex embodied AI tasks, especially when models trained on real images are used in simulation (eg. Mask-RCNN trained on MS-COCO). We also expect models trained on HM3D to transfer better to the real world.
>
> ### Q7 - How to count the area of stairs in multi-floor buildings? Are the robots capable of going up and down the stairs?
> Stair regions are approximated by a “flat ramp” in the navigation mesh, which we use directly as an approximation of the area covered by the stairs.  The agents in simulation are capable of going up and down the stairs by navigating up this “ramp”. See supplementary video 00337-CFVBbU9Rsyb/nav.mp4. For real robot experiments, it will require legged robots such as Spot by Boston Dynamics or AlienGo by Unitree to go up and down the stairs.

---

> > ### Author Response · Authors · 2021-09-28
> > **Response to reviewer HSHf [Part 2]**
> >
> > ### Q8 - L217 - when evaluate agents (trained on different datasets) on a test environment, does the agents start from the same random locations to the same target goal?
> > Yes. This is defined by the PointNav dataset. For each scene in MP3D, Gibson, HM3D, there is a corresponding PointNav dataset that contains a predefined set of (start, end) locations. All agents are evaluated on the same episodes.
> >
> > ### Q9 - L224 - what is SPL?
> > SPL stands for “Success weighted by (normalized inverse) Path Length”. It weighs episode success based on how efficient the navigation was relative to the shortest path trajectory from start to finish. See Section 4 recommendations 2 & 3 from https://arxiv.org/pdf/1807.06757.pdf for the definition.
> >
> > ### Q10 - L230 - the training episodes are 4.11M for Gibson and 8M for HM3D. Is the comparison fair?
> > It is fair in the sense that we generate an equal number of episodes (10,000) per scene (L231 footnote). Note that sampling more episodes per scene is unlikely to help since the underlying scene remains the same, and the episode diversity will not improve.
> >
> > ### Q11 - Fig 8 - top left subfigure - why are Gibson agents better than HM3D agents on Gibson (test)?
> > The performance of the Gibson agents and the HM3D agents is within the standard error margins, thus the difference is not statistically significant. The Gibson agent scores 0.94 +- 0.1 SPL and the HM3D agent scores 0.93 +- 0.1 SPL (see Table 2).
> >
> > ### Q12 - how does this dataset compared to ARKitScenes?
> > As discussed in L87-9, ARKitScenes consists of mostly single room-scale reconstructions acquired from rentals in 3 European cities. HM3D consists of 1,000 building-scale reconstructions spanning 38 countries. Unfortunately, the dataset is not yet publicly available so we cannot yet provide a more detailed  comparison (e.g., Table 1, Figures 4 & 5).
> >
> > ### Q13 - missing reference on OpenRooms (https://arxiv.org/abs/2007.12868v2)
> > Thank you for the reference. We have cited it in L78 along with relevant references.

---

> > > ### Comment · Reviewer_HSHf · 2021-09-30
> > > **reply to author**
> > >
> > > Thanks for the rebuttal. I am aware ARKitScenes is concurrent work, just point out to see your opinion of its data quality vs yours. I believe HM3D dataset is useful for the embodied AI community. Also as disclaimer, I am not related to OpenRooms dataset. It would be nice to see the raw scanner data (similar to ScanNet or MP3D) and semantics labels to be released in the future, for more 3DRU tasks. I will keep my rating.

---

> ### Comment · Reviewer_VcTt · 2021-09-28
> **Comments from Reviewer VcTt**
>
> Hi, I agree with most of your reviews. But I'd like to point out that ARKitScenes is submitted to NeurIPS 2021 Dataset Track round 1. I think HM3D and ARKitScenes are considered concurrent works.
>
> See https://openreview.net/forum?id=tjZjv_qh_CE

---

### Official Review · Reviewer_VcTt · 2021-09-20

**Rating:** 8
**Confidence:** 4
**Correctness:** The dataset is constructed in a sound…
**Clarity:** Yes

**Strengths:**

1. HM3D emphasizes both the visual fidelity and the reconstruction completeness. As a real 3D scan dataset, this is very important to a lot of downstream tasks. For example, bad visual fidelity will limit the generalization from COCO pretrained model to the target dataset, since the model has never "seen" these images. Replica has good visual fidelity but it is a small dataset. The great visual and reconstrution quailty of HM3D will definitely advance the progressive of embodies AI.

2. The experiments are extensive. The navigation agent is trained and cross-validated on Gibson, MP3D and HM3D. It demonstrates the effectiveness of HM3D.

3. The data collection is diversified in terms of geographic regions. As stated in the paper, scenes come from 38 countries and 181 geographic regions. This will help models generalize better.

**Weaknesses:**

1. The dataset is only validated in pointgoal navigation. However, HM3D is actually a very useful dataset and many tasks can be trained and evaluated on HM3D. Real-world 3D datasets are very useful and important for 3D computer vision systems such as single view depth estimation. Besides that, embodies ai also has other tasks besides pointgoal navigation. By looking at table 2, I feel it's actually an easy task, especially when the input has depth. An agent trained on MP3D has already had 0.90 SPL on Gibson dataset. Given that the improvement is mainly on hard episodes, evaluating on a more challenging task makes more sense to me.

2. The dataset does not specify all the available ground truth. From the experiments I believe 3D mesh, depth and rgb are available. However, how about other annotations? For example, floors and rooms are very useful and supported in AI Habitat. Gibson does not have it while MP3D has it. Is it available in HM3D? Also, does it have semantic annotations (most datasets in Table 1 have and it's useful)?

**Additional Feedback:**

> L140 This is computed as the ratio between the raw scene mesh area within 0.5m of the navigable regions and the navigable space

According to my understanding, it is the ratio between all the raw scene mesh area and navigable space. all mesh area = navigable space + obstacles. Is it correct? Why is 0.5m necessary here?

> Fig 5(a)

The camera pose of real and rendered images do not look the same. Why?

>  Finally, the visual fidelity of images rendered from HM3D is 20 - 85% higher than prior large-scale datasets,

How do you compute 20 - 85%? Is it based on FID? I cannot compute it from Fig 5(b).

**Documentation:**

Details about data collection are available in supp material. It is publicly available now for academic usage.

**Ethics:**

I cannot foresee any ethical concerns.

**Relation To Prior Work:**

Yes. Extensive comparison is conducted in the paper including MP3D, Gibson and ScanNet.

**Summary And Contributions:**

The paper introduces HM3D, which is a large scale 3D scan dataset for embodied AI. It demonstrates the scale and quality of the dataset, both visually and numerically. Extensive experiments about PointGoal navigation are conducted in the dataset, including comparison with Gibson and MP3D.

---

> ### Author Response · Authors · 2021-09-28
> **Response to reviewer VcTt**
>
> ### Q1 - dataset is only validated in pointgoal navigation … easy task … evaluating on a more challenging task
> We agree with the reviewer. We benchmark HM3D on PointNav since it is a fundamental task that can serve as a “building block” for other embodied AI tasks, thus making it a good starting point (L215-217). Our evaluation on PointNav revealed that the diversity, scale, and fidelity of HM3D can benefit embodied tasks. As the reviewer states, many other tasks can already be defined on HM3D (eg. ImageNav, Exploration, ObjectNav with inserted objects), and we plan to add support for more semantic tasks in the future. Please see response Q3 to R-Lq43 for examples of tasks that are supported in HM3D.
>
>
> ### Q2 - dataset does not specify all the available ground truth (floors, rooms, semantics)
> The number of floors and rooms are released as a part of the metadata (see our website https://aihabitat.org/datasets/hm3d/). While we do not include semantics, we are actively working on augmenting the dataset with semantics (as acknowledged in Supplementary Section S1).
>
> ### Q3 - scene clutter - why is 0.5m necessary?
> The complete mesh area includes outer surface areas of the scenes (i.e. outside the building) and other inaccessible parts of the building. We restrict to 0.5m to only pick the surfaces that are near navigable spaces in the building (e.g., furniture, and interior walls). Due to the lack of semantic annotations, we’re assuming that surfaces within 0.5m of a navigable location are more likely to correspond to clutter than surfaces further away from all navigable locations.
>
>
> ### Q4 - Fig 5 (a) - camera pose of real and rendered images do not look the same
> The figure is meant purely for illustration purposes. We approximately matched the pose of real images by manually navigating an agent within the simulator. We have clarified this in the paper.
>
> ### Q5 - How do you compute 20-85 % higher visual fidelity?
> This is based on the KID scores in Figure 5(b). The formula is:
>
> $$
> 100 \times \frac{mKID(X) - mKID(HM3D)}{mKID(X)}
> $$
>
>  where $mKID$ is the mean KID from Gibson real and MP3D real, and $X$ can be Replica ($18.6$%), Gibson ($48.4$%), MP3D ($43.2$%), or ScanNet ($85.9$%). We have clarified this in the paper.

---

> > ### Comment · Reviewer_VcTt · 2021-09-28
> > **Response to Authors**
> >
> > Thank you for your clarification. Q2-5 make sense to me. For Q1, I still believe HM3D can be beneficial to more 3D vision tasks other than navigation. I'll keep my rating.

---

### Official Review · Reviewer_Lq43 · 2021-09-20
**A high-quality dataset but seems to be simple**

**Rating:** 6
**Confidence:** 3
**Correctness:** Yes.
**Clarity:** Yes, the paper is well written and ea…

**Strengths:**

- The proposed dataset is on a larger scale and the reconstruction quality is quite better than previous datasets.
- The comparison with previous datasets is clear with lots of statistical and experimental support.
- The transferring experiment of the PointGoal navigation task shows the better quality of the proposed dataset over the previous.
- The dataset is publically accessible with good documentation. The previewing website is cool and we can easilly have a brief view of each scene.

**Weaknesses:**

- The proposed dataset seems to be simple as the algorithms trained and tested on it achieve almost 100% (99%Depth & 98%RGB) performance in Table 2. That limits its application and promotion to the community as it didn't reveal any new challenging problems that will guide and help to boost the development of the field. For the example PointNav task shown in the paper, the proposed dataset is easier than the previous MP3D, which limits the upper bound of algorithms developed and selected from it.
- The paper simply manually removed those scenes that failed to be reconstructed, which may hurt the distribution between the selected dataset and the real world and lead to a simpler dataset. What caused the reconstruction failure? The depth missing because of out of range of depth sensors or objects with reflective, transparent, or dark surfaces that cannot be captured by depth sensors? And are these challenging cases what agents with RGBD sensors confront in the real world? It's thankful to have an analysis and give some insight into the field.
- The paper didn't compare their reconstruction procedure with previous datasets, which is also helpful to evaluate the dataset quality.
- If the dataset is labeled with object semantics and physical attributes, that will enable more application and get more impact.

**Additional Feedback:**

None.

**Documentation:**

The data is collected by Matterport Pro2 users and post-processed by the authors. However, some technical details of post-processing are missing. For example, what kind of technique is used to reconstruct the scene and its comparison to those used by previous datasets. Providing this information will help evaluate the dataset quality.

**Ethics:**

No.

**Relation To Prior Work:**

Yes.

**Summary And Contributions:**

The paper introduced a large-scale dataset of 3D Environments for Embodied AI. The proposed dataset is with a larger scale, more complete of reconstructed scenes, and higher visual fidelity over previous datasets. The paper also benchmarks the PointGoal navigation task on the proposed dataset and existing ones and shows the better transferring capability of agents trained on the proposed HM3D over others.

---

> ### Author Response · Authors · 2021-09-28
> **Response to reviewer Lq43**
>
> ### Q1 - proposed (PointNav) dataset seems to be simple ... limits its application
> We generally agree with R-Lq43’s observations. However, we do want to distinguish between the HM3D dataset of 3D scenes vs. the HM3D-PointNav dataset. The PointNav dataset contains episodes for PointGoal Navigation in HM3D with a specific “v1” configuration of the agent (matching Habitat challenge 2019). What R-Lq43 is calling “simple” (which we agree with) is the PointNav-v1 dataset. For the same HM3D scenes, we can construct the harder PointNav-v2 dataset (matching Habitat challenge 2021 config) and, most importantly, create episode datasets for other tasks such (like ImageNav, Exploration, ObjectNav with inserted objects, Pathdreamer-style image generation, novel view synthesis, etc). Thus, the simplicity of the HM3D PointNav-v1 dataset does not limit the application of the HM3D scenes dataset since PointNav is not the only task that HM3D supports (see response Q3 for other tasks that HM3D supports).
> We experiment with PointNav since it is a fundamental task that can serve as a “building block” for other embodied AI tasks, thus making it a good starting point (L215-217).  The difficulty for PointNav is largely determined by the geodesic distance between the start and goal locations (L264-5). MP3D scenes have a higher navigable area per scene compared to HM3D (~$335 m^2$ in MP3D vs. ~$112.5 m^2$ in HM3D, see Table 1). Thus, MP3D is likely to have more difficult PointNav episodes than HM3D.
>
>
> ### Q2 - didn’t compare reconstruction procedure with previous dataset
> As we describe in Section 3, L102-104,  the reconstruction procedure used is proprietary to Matterport Inc., and is not publicly released. Note that the contribution of this paper is the reconstructed scenes themselves, not the reconstruction procedure for obtaining these scenes.
>
>
> ### Q3 - if dataset is labeled with object semantics and physical attributes, … more impact
> Agreed. Having semantics is definitely valuable, and we are working in that direction. However, even without semantics, the scale, diversity, and fidelity of HM3D scenes are valuable for embodied AI research (as we show through the experiments in the paper). HM3D already supports more complex navigation tasks without requiring additional annotations. We list some examples below.
> **1. PointNav-v2:** It upgrades the PointNav-v1 task by removing the GPS+Compass sensor which simplified the localization problem, and introduces RGB-D sensor noise and actuation noise. Thus, this task is supported in HM3D. Reference: https://aihabitat.org/challenge/2021/
> **2. ImageNav:** The navigation target is specified as an RGB image of the location to reach (eg. picture of the living room) instead of an (x, y) coordinate. This task is supported since we only require the ability to render RGB images and compute shortest path trajectories (computed via the NavMesh construction in Habitat-sim, see https://aihabitat.org/tutorial/2020/). Reference: Chaplot et al., Neural Topological SLAM for Visual Navigation, CVPR 2020
> **3. Exploration:** The navigation objective is to cover as much area as possible within a restricted time budget in a novel environment. This task is supported since it only requires depth sensors + camera poses to construct maps, which can then be used to compute the area seen. Reference: Chen et al., Learning Exploration Policies for Navigation, ICLR 2019
> **4. ObjectNav with inserted objects:** New objects of various categories such as hand tools, fruits, etc., are inserted into an existing 3D scene. The navigation goal is to find these objects given their categories. This task is supported since it only requires 3D models of objects (available via datasets such as https://www.ycbbenchmarks.com/), and the ability to insert an object model into a 3D scene (supported via habitat-sim https://aihabitat.org/docs/habitat-sim/rigid-object-tutorial.html).

---

> > ### Comment · Reviewer_Lq43 · 2021-09-29
> > **Response to Authors**
> >
> > Thanks for the clarification.
> >
> > For Q1, I agree that HM3D can be applied to more tasks. For the simple dataset concern, though PointNav is a fundamental task for embodied AI, the result in table 2 doesn't show the strength of HM3D over existing ones but reveals its weakness in complexity compared to MP3D. Specifically, as also mentioned by reviewer HSHf, HM3D uses 8M train episodes while Gibson uses 4.11M, the improvement may come from more training data from more scenes, given that the larger Gibson also surpasses Gibson 4+ though the latter is with higher quality. I think it's not so convincing to say that the main attributes of HM3D (larger scene scale, more complete, higher fidelity) contributes to the improvement. Maybe PointNav is an easy task and benchmark on more challenging tasks that can prove the strength of HM3D makes more sense.
> >
> > For Q2, it seems the pipeline has some improvement over existing ones in the response to Reviewer HSHf, which addresses my concern about algorithm comparison. But I still believe a brief discussion on the removed failures cases showing the limitation of current works is beneficial to the community, for example, the sim-to-real transfer concern.
> >
> > For Q3, happy to see that more labels are in progress and more tasks are supported.
> >
> > Overall, since more labels are in progress and more tasks will be supported in the ai habitat challenge, I would raise my score.

---

> > > ### Author Response · Authors · 2021-09-29
> > > **Response to reviewer Lq43 --- discussion on rejection criteria for reconstructions**
> > >
> > > Thank you for raising this issue. We rejected scenes if they were incomplete, had significant reconstruction artifacts, or potential privacy concerns. Specifically, we rejected scenes which had:
> > > 1. Large holes/cracks in the floors, walls, or staircases which lead to separated “islands” within a single scene. These impact navigability between different parts of the scene for embodied agents in simulation.
> > > 2. Large holes/cracks on object surfaces which can impact perception as well as potentially hinder future semantic annotations.
> > > 3. Reconstruction artifacts/errors (i.e. bogus meshes) due to reflections and lighting. The scanning process occasionally misinterpreted unusual lighting/shadows as objects.
> > > 4. Large outdoor areas in which many parts were out of range of the depth sensors.
> > > 5. Uninteresting content with limited to no interactive potential (e.g., large empty parking lots, basement storage areas, abandoned warehouses, empty halls, hazardous waste store room).
> > > 6. Objects that changed positions in the middle of the scanning process. For example, the door to a room may be open during a part of the scanning process, and closed for the latter part. This gave rise to artificial debris in the room entrance.
> > > 7. Many doors that were scanned partially or fully closed (affects navigability between rooms).
> > > 8. Brand names which were clearly visible (usually commercial buildings).
> > > 9. Humans that were partially or fully visible in the scan.
> > > 10. Other scenes which were naturally hard to scan and had poor reconstruction quality (e.g., submarine and ship exhibits, open-pit mines with equipment like excavators).
> > >
> > > We agree that these choices may impact the distribution over real scenes since some artifacts are correlated with either challenging surface materials or lighting conditions in the real world (especially points 1-4). However, we chose to focus on interiors, and tried to minimize reconstruction issues that affect navigability in simulation. Particularly, we prioritized surface completeness and the absence of spurious geometric artifacts as these have the biggest impact on RGB-D rendering in simulation (as we note in Sec. 3.2). As requested, we will add a brief discussion on some of the rejected failure cases highlighting the limitations of current reconstruction pipelines. We hope that this will be beneficial to the community.

---

### Author Response · Authors · 2021-09-28
**Common response**

We thank the reviewers for the insightful comments and questions. The reviewers appreciate that HM3D has larger scale (R-Lq43, R-HSHf), higher reconstruction quality (R-Lq43, R-VcTt), higher visual fidelity / perceptual quality (R-VcTt, R-HSHf), high geographical diversity (R-VcTt, R-HSHf), and that the dataset is easily accessible (R-Lq43). The reviewers also felt that the experiments and statistical comparisons are extensive (R-VcTt) and clear (R-Lq43). We respond to specific queries below and incorporate all feedback in the paper.

---

### Decision · Program_Chairs · 2021-10-09

**Decision:**

Accept

**Comment:**

All the reviewers agree that this is a high-quality dataset, easily accessible, and larger-scale compared to existing datasets. Several reviewers noted that the PointNav task showcased is close to saturation, hence this benchmark will not drive much progress on this task. However, other tasks can be set up, as the authors noted, hence this is still a very useful resource for the Computer Vision and Robotics communities.